# Therapeutic Strategies for Targeting Ovarian Cancer Stem Cells

**DOI:** 10.3390/ijms22105059

**Published:** 2021-05-11

**Authors:** Wookyeom Yang, Dasol Kim, Dae Kyoung Kim, Kyung Un Choi, Dong Soo Suh, Jae Ho Kim

**Affiliations:** 1Convergence Stem Cell Research Center, Pusan National University, Yangsan 50612, Gyeongsangnam-do, Korea; wookyeom@pusan.ac.kr; 2Department of Physiology, School of Medicine, Pusan National University, Yangsan 50612, Gyeongsangnam-do, Korea; dasolk95@naver.com (D.K.); kyumkiki@gmail.com (D.K.K.); 3Department of Pathology, Pusan National University Hospital, Busan 49241, Korea; kuchoi@pusan.ac.kr; 4Department of Obstetrics and Gynecology, Pusan National University Hospital, Busan 49241, Korea; dssuh@pusan.ac.kr

**Keywords:** cancer stem cells, ovarian cancer, chemoreistance, stemess

## Abstract

Ovarian cancer is a fatal gynecological malignancy. Although first-line chemotherapy and surgical operation are effective treatments for ovarian cancer, its clinical management remains a challenge owing to intrinsic or acquired drug resistance and relapse at local or distal lesions. Cancer stem cells (CSCs) are a small subpopulation of cells inside tumor tissues, and they can self-renew and differentiate. CSCs are responsible for the cancer malignancy involved in relapses as well as resistance to chemotherapy and radiation. These malignant properties of CSCs are regulated by cell surface receptors and intracellular pluripotency-associated factors triggered by internal or external stimuli from the tumor microenvironment. The malignancy of CSCs can be attenuated by individual or combined restraining of cell surface receptors and intracellular pluripotency-associated factors. Therefore, targeted therapy against CSCs is a feasible therapeutic tool against ovarian cancer. In this paper, we review the prominent roles of cell surface receptors and intracellular pluripotency-associated factors in mediating the stemness and malignancy of ovarian CSCs.

## 1. Introduction

Ovarian cancer has the highest mortality rate among gynecological cancers. The World Cancer Report of the International Agency for Research on Cancer stated that 295,414 women were diagnosed with ovarian cancer in 2018 [1]. Diagnosis at an early stage has numerous benefits such as a longer survival rate; however, 80% of ovarian cancer patients are diagnosed at an advanced stage (III or IV) because they exhibit no symptoms in the early stages, following which the tumor disseminates to the peritoneal cavity [2,3,4]. Ovarian cancer has high heterogeneity, which is generated by various factors, including different histopathological properties, origin, clinical evolution, response to treatment, and genomic alteration, and can be divided into nine subtypes: serous, endometrioid, clear cell, mucinous, Brenner tumor, transitional cell, squamous cell, mixed epithelial, and undifferentiated [5]. Standard ovarian cancer treatment comprises first-line chemotherapy with a carboplatin-paclitaxel regimen and surgical debulking [4,6]. Although this regimen is initially effective, recurrence is persistent owing to intrinsic or acquired chemoresistance [7,8]. Most ovarian cancer patients suffer at least one relapse within 12 to 18 months [9,10] owing to chemoresistance and metastasis [3,11]; hence, 70% of these patients die within five years [1,4,9]. Therefore, ovarian cancer treatment urgently needs to address challenges such as recurrence, chemoresistance, and metastasis.

Cancer stem cells (CSCs) are highly tumorigenic as tumor-initiating cells and possess the ability to self-renew, similar to normal stem cells [12,13]. Although CSCs are a rare population of tumor cells depending on the tumor type and stage [1], they are responsible for drug resistance, recurrence, and metastasis in response to tumor microenvironmental or non-microenvironmental stimuli [14,15,16,17]. Despite contradictory findings, the upregulated CSC pathway is responsible for the poor outcomes of many aggressive tumors [18]. CSC-related cellular markers comprise cell surface and intracellular markers, which are closely connected via a complicated pathway that interplays with the tumor microenvironment. Many reports suggest that the individual or combined inhibition of CSC marker expression or activity effectively diminishes the growth and dissemination properties of CSCs both in vitro and in vivo [19,20].

After CSCs were first found in ovarian cancer patient ascites [21], many studies have investigated the correlation between CSCs and ovarian cancer. Accumulating evidence indicates that the elimination of CSCs is required to inhibit ovarian cancer growth and frequent relapse [19,20]. Many studies have shown that there are stem-like epithelial cells in the ovarian surface [22] and fallopian tube epithelium [23]. Stem cells have been identified in the coelomic epithelium of mouse ovaries [24] and in a transitional region known as the hilum [25]. Although there is some evidence to the contrary, ovarian cancer is considered to originate from both these tissues [20,26]. Moreover, stem cell factors increase sphere formation and tumor transformation potential by inactivating TP53 and Rb1 via mutations; two tumor-suppressor genes are frequently mutated in high-grade serous ovarian cancer [27]. Stem cells play a crucial role in maintaining normal tissues, but they may be a double-edged sword because dysregulation and transformation through the mutation of tumor suppressors can cause cancer [28] (Figure 1). In this paper, we review potential therapeutic or diagnostic biomarkers for CSCs and discuss recent studies on approaches for targeting ovarian CSCs.

## 2. Cell Surface Markers of Ovarian CSCs

### 2.1. CD44

CD44 is a surface transmembrane glycoprotein that acts as a receptor, participating in various physiological processes such as cell–cell interaction, cell migration, and cell–matrix adhesion [29]. CD44 is one of the most common CSC markers, which is used individually or in combination with other potential markers to verify CSCs [30,31,32], and it is highly expressed in several tumors such as those of the ovary, breast, and pancreas [33,34,35]. CD44+ cells have been identified in primary tissues, metastatic tissues, and malignant ascites of ovarian cancer [36]; one hundred CD44+/CD117+ cells are sufficient to propagate the original tumor, but CD44-/CD117- cells are not [30]. CD44 increases invasion and migration activity in non-malignant ovarian cancer cells via exosome transfer [37]. CD44 elevates chemoresistance and invasion activity through a signal transducer and activator of transcription 3 (STAT3)-dependent mechanism, and it influences angiogenesis and immunosuppression in the tumor microenvironment via the secretion of various angiogenic factors and cytokines, including vascular endothelial growth factor and interleukin 6 [38]. Despite the function of CD44 as a stem cell biomarker, contradictory findings suggest that CD44 fails to function as a prognostic factor in ovarian cancer [39,40]. Nevertheless, CD44 can be utilized as an outstanding prognostic marker in combination with other putative biomarkers [41,42].

### 2.2. CD117

CD117, generally known as c-kit, is a receptor tyrosine kinase that is involved in various signaling pathways responsible for the survival, proliferation, migration, tumor progression, and stemness [19,43] of CSCs, and it is a common marker of hematopoietic stem cells, mesenchymal stem cells (MSCs), and embryonic stem cells [44]. CD117, in combination with CD44, has been used to identify CSCs [30]. CD117+ ovarian cancer cells overexpress SOX2, octamer-binding transcription factor 4 (OCT4), and NANOG, which are CSC markers involved in stemness and chemoresistance [45]. CD117 overexpression in ovarian CSCs elevates tumor-initiating capacity and chemoresistance against cisplatin/paclitaxel via the induction of the Wnt/beta-catenin-ATP–Binding Cassette Subfamily G Member 2 (ABCG2) axis [46]. In contrast, miRNA-26b is under-regulated in human CD117+CD44+ ovarian CSCs. miRNA-26b overexpression inhibits cell proliferation and promotes cell apoptosis [47]. According to a meta-analysis, CD117 expression significantly correlates with patient age, tumor differentiation grade, advanced stage, and histological type, and CD117 overexpression is associated with poor overall survival in ovarian cancer [48].

### 2.3. CD24

CD24, which encodes a glycosylphosphatidylinositol-linked cell surface ligand for P-selectin, is closely associated with serous ovarian cancer [49]. CD24 has been identified as one of the most important CSC markers in several cancers, including ovarian and colorectal cancers [50,51]. CD24+ cells from ovarian tumor specimens regulate proliferation, self-renewal and differentiation, chemoresistance, and tumorigenicity, which are CSC characteristics [50]. CD24+ cells could recapitulate the development of the original tumor in a mouse model of ovarian cancer with the conditional deletion of Apc, Pten, and cellular tumor antigen p53 (Trp53; *Mus musculus*), which was confirmed through JAK2–STAT3 signaling [52]. CD24 regulates epithelial-to-mesenchymal transition (EMT) by modulating the PI3K/AKT/MAPK signaling pathway [53]. The overexpression of CD24 has been reported in ovarian cancer patients, and it is a metastatic prognosis marker for poor survival [53,54]. Triple-positive (CD24+/CD44+/EpCAM+) cells isolated from ovarian cancer patients exhibit clonogenic potential and chemoresistance to cisplatin and doxorubicin [55]. However, the CD44+/CD24- phenotype in ovarian cancer cells determines the CSC properties of newly propagated and invasive tumors, and ovarian cancer patients with this phenotype exhibit enhanced recurrence and shorter progression-free survival [56]. Therefore, CD24 should be used in combination with other stem cell markers, such as CD44 and epithelial cell adhesion molecule (EpCAM), for the isolation of ovarian CSCs.

### 2.4. CD133

There is evidence that CD133, a glycosylated transmembrane protein, helps maintain cancer stemness and is associated with tumor metastasis [57]. CD133 has been reported as a prognostic marker and regulator of cancer metastasis in several cancers, such as ovarian cancer, glioblastoma, and prostate cancer [58,59,60]. Primary cancer tissues comprise 0.1% to 3% CD133+ cells; however, the proportion of CD133+ cells increases upon chemotherapy with cisplatin or paclitaxel [61]. The expression level of CD133 is elevated in sphere-forming and drug-resistant populations of ovarian cancer cells, which exhibit chemoresistance and tumorigenesis in vivo and increased levels of stemness markers such as OCT4, SOX2, and NANOG [62]. CD133+ cells promote non-stem cancer cell metastasis by inducing EMT via the CCL5–NF-κB axis [63]. Moreover, CD133 expression significantly correlates with a low survival rate in ovarian cancer [64]. Based on the aforementioned results, CD133 is a promising prognosis marker and therapeutic target for ovarian CSCs.

### 2.5. CD166/ALCAM

CD166, also known as activated leukocyte cell adhesion molecule (ALCAM), is a transmembrane glycoprotein of the immunoglobulin superfamily [65] and is expressed mainly at the cell membrane [66]. CD166 is overexpressed in various cancers, including thyroid, head and neck, lung, and liver cancers [67]. A high level of CD166 is associated with a poor prognosis in malignant melanoma, the metastasis of prostate cancer, the invasion activity of cholangiocarcinoma, and the anti-apoptotic function of breast cancer [67,68,69,70]. Cells expressing high levels of CD44/CD166 exhibit enhanced CSC-like properties and tumorigenicity. CD166 is overexpressed in tissues from recurrent tumors and is associated with poor prognosis in head and neck squamous cell carcinoma [71]. In addition, CD166 regulates the expression of CSC markers and mediates the EGFR/ERK1/2 pathway in nasopharyngeal carcinoma [72]. CD166 has been reported to play a pro-carcinogenic role in liver cancer cells by promoting the expression and activation of RAC-alpha serine/threonine-protein kinase (AKT) and yes-associated protein (YAP), which is a coactivator of Hippo signaling [73,74]. Our previous study showed that CD166 exhibits CSC-like properties in primary epithelial ovarian cancer cells. CD166 induced the expression of CSC markers such as OCT4, SOX2, and aldehyde dehydrogenase 1 A1 (ALDH1A1), and ABC transporters in both A2780-derived sphere-forming cells and primary ovarian CSCs, thus promoting CSC-like properties and chemoresistance [75]. A recent report indicated that CD133+/CD166+ cells are strongly stem-like cancer cells in human gastric adenocarcinoma, and that CD133+/CD166- cells exhibit self-renewal properties, colony formation capacity, and substantial migration activity [76]. These results suggest that CD166 may be a potential therapeutic target for CSCs, including ovarian CSCs.

## 3. Intracellular Markers of Ovarian CSCs

### 3.1. Aldehyde dehydrogenase 1

Aldehyde dehydrogenase (ALDH) is an enzyme that catalyzes aldehydes, which detoxify endogenous and exogenous reactive aldehydes [77]. ALDH comprises three classes: class 1 (cytosolic), class 2 (mitochondrial), and class 3 [78]. ALDH1 consists of three isozymes (ALDH1A1, ALDH1A2, and ALDH1A3); ALDH1A1 predominantly serves as a CSC marker. ALDH1 expression in ovarian cancer cells has been demonstrated in several studies [79,80]. Our previous study and others suggest that spheroid cells derived from ovarian cancer cell lines and primary ovarian cancer tissues are enriched with CSC-like cells exhibiting high ALDH activity, elevated stem-cell marker expression, self-renewal, high proliferation, and differentiation potential [81,82,83]. The ALDH1A1 subpopulation is associated with an invasive phenotype, clonogenicity, drug resistance, and worse progression-free survival in ovarian cancer patients [84,85]. Accumulating evidence suggests that ALDH1A1 regulates the maintenance of ovarian CSCs, ALDH+ ovarian cancer cells exhibit stem cell-like properties, and knock-down of ALDH1A1 in ovarian cancer cells diminishes clonogenic ability [85].

### 3.2. Autotaxin

Autotaxin (ATX), which belongs to the ectonucleotide pyrophosphatase/phosphodiesterase family, is a tumor cell motility-stimulating factor that stimulates cell motility and cell proliferation of cancer cell lines [86]. ATX exhibits lysophospholipase D activity, and it affects tumor motility and growth by producing lysophosphatidic acid (LPA) [86,87]. Several studies have shown that ATX is expressed in several tissues such as the ovary, small intestine, placenta, platelets, and adipose tissue, and body fluids such as cerebrospinal fluid [88,89]. A high expression of ATX has been detected in breast cancer, glioblastoma multiforme, prostate cancer, hepatocellular carcinoma, and melanoma [90,91,92,93,94]. ATX overexpression in these cancers promotes tumor motility and invasiveness, enhances metastatic potential, and correlates with poor outcomes in cancer patients [95]. It has been reported that ATX is responsible for maintaining ovarian CSCs via the LPA–LPAR axis. ATX-induced LPA production in A2780 sphere-forming cells enhances migration, sphere formation, and the expression of CSC markers, such as OCT4, SOX2, Kruppel-like factor 4 (KLF4), and ALDH1. Conversely, silencing ATX expression in A2780-derived spheroid cells reduces the level of stemness-related transcription factors such as OCT4, SOX2, and KLF4 [83]. Therefore, the ATX–LPA signaling axis may be a prominent target for the development of combination therapies for ovarian CSCs.

### 3.3. Pluripotency-Associated Factors

The pluripotent properties of CSCs, such as long-term self-renewal, multi-differentiation potential, and asymmetric division [12,96], have been attributed to common stem-related transcription factors [97]. They are derived from the embryonic transcription factors OCT4, SOX2, and NANOG, whose genes are considered stem cell signature genes [97]. These gene signatures with c-MYC (myc proto-oncogene protein) are frequently overexpressed in poorly differentiated tumors and then well differentiated in breast cancer, glioblastoma, and bladder carcinomas. The expression signatures of OCT4, SOX2, and NANOG are associated with poor clinical outcomes in breast cancer [98]. Green fluorescent protein (GFP)-labeled ovarian CSCs utilize the NANOG promoter system to overexpress OCT4, SOX2, and NANOG compared with the GFP-negative ovarian cancer cells, and the GFP-positive cells also exhibit greater cisplatin resistance and tumor initiation properties. In addition, low-dose cisplatin treatment induces stemness in ovarian cancer cells [99]. OCT4, SOX2, and NANOG are significantly overexpressed in high-grade serous ovarian cancer cell lines cultured under 3D culture conditions, and CD117+ or ALDH+/CD133+ cells exhibit elevated expression of stemness genes [45]. SOX2, but not OCT4 or NANOG, has been implicated in early tumor initiation and plays a more substantial role in tumor relapse in ovarian cancer patients. SOX2 expression is upregulated in several types of CSCs, including those of breast, gastric, lung, and ovarian cancers [100]. SOX2 with ALDH and ABC transporters is overexpressed under hypoxic conditions via neurogenic locus notch homolog protein 1 (NOTCH1) in ovarian CSC spheroid culture. A knockdown of SOX2 expression inhibits ovarian CSC survival and disturbs spheroid formation [101]. However, the precise regulatory mechanism of these pluripotent factors in ovarian CSCs is not understood.

### 3.4. Hypoxia-inducible Factor 1-alpha (HIF-1α)

Hypoxia refers to a state of insufficient oxygen supply caused by various factors, including physiological and physical states, epigenetic environments, and gene alterations [102,103]. Hypoxia is an essential hallmark of cancer cells and their microenvironment, and it confers an advantage to cancer cells during their growth, survival, and metastasis [103]. Therefore, it is important to control the hypoxic microenvironment to inhibit the malignant properties of cancer. HIF-1α, one of the master regulators of hypoxia adaptation, is associated with angiogenesis, glucose metabolism, cell proliferation, and drug resistance [104]. HIF-1α can induce pluripotent stem cell inducers, such as OCT4, NANOG, SOX2, KLF4, c-MYC, and microRNA-302, in 11 cancer cell lines (derived from prostate, brain, kidney, cervix, lung, colon, liver, and breast cancers) [105]. Hypoxia plays an essential role in maintaining CSC characteristics, such as colony and sphere formation, and the expression of CD133 and CD44 in ovarian CSCs [106]. CD166+ lung adenocarcinoma A549 cells showed robust drug resistance and stemness owing to HIF-1α-induced ABCG2 under chemically induced hypoxia conditions [107]. Our previous study showed that hypoxia-induced NOTCH and HIF-1α elevated SOX2 expression to promote sphere formation and drug resistance by increasing ALDH1, ATP-Binding Cassette Subfamily B Member 1 (ABCB1), and ABCG2 expression in ovarian cancer cells. In contrast, silencing of SOX2 in A2780-derived ovarian CSCs significantly reduced stem cell marker expression and decreased resistance to chemotherapy drugs such as doxorubicin or paclitaxel, as in ovarian cancer patient-derived sphere cells [101]. These reports suggest that controlling hypoxia signaling may be novel options to eliminate CSCs in ovarian cancer. The summary of ovarian stem cell markers and their features as mentioned above was compiled on Table 1 with the references.

## 4. Therapeutic Strategies Using CSC Markers

### 4.1. Standard and Targeted Therapies for Ovarian Cancer

Ovarian cancer patients show a high response rate to front-line chemotherapy based on platinum-based drugs with surgical debulking [6], but they eventually relapse within several months [9,10]. Platinum resistance is defined as the development of recurrence within six months following first-line chemotherapy [2]. Platinum-resistant patients have been administered combined chemotherapy with or without doxorubicin, paclitaxel, gemcitabine, or targeted therapies such as the administration of bevacizumab or poly-ADP ribose polymerase (PARP) inhibitors [108]. In 2018, a phase 3 clinical trial showed that olaparib, a potent PARP inhibitor, dramatically improved the disease-free survival of ovarian cancer patients carrying the breast cancer type susceptibility protein 1/2 (BRCA1/2) mutation by 70% (3 years) compared with the placebo [109,110]. However, BRCA wild-type ovarian cancer patients showed a lower response rate to PARP inhibitors [109,111]. PARP inhibitor resistance and recurrence have been reported in ovarian [112] and breast cancers [113]. Combined therapy with other targeted therapeutic drugs, including PI3K, cyclin-dependent kinase 1 (Cdk1), and hepatocyte growth factor receptor (c-MET) inhibitors, was attempted to overcome PARP inhibitor resistance [1]. PARP inhibitor treatment induced cell apoptosis in non-ovarian CSCs but enriched CD133+ and CD117+ ovarian CSCs by increasing DNA repair ability in a BRCA mutation-independent manner [114], suggesting that ovarian CSCs may exhibit intrinsic or acquired resistance against PARP inhibitor treatment, leading to relapse. Therefore, it is essential to eliminate ovarian CSCs from ovarian cancer patients.

CSC surface markers are more accessible to therapeutic antibodies and small molecules than other CSC markers; thus, it is a feasible target that can be utilized to treat various cancers [115]. Moreover, considering the function of CSC markers in maintaining CSC properties, it is straightforward to isolate specific CSCs and use the diagnostic markers for predicting patient survival. Although there are several debates on the isolation and targeting of CSC surface markers, many researchers believe that eliminating specific CSCs can eradicate whole cancer tissues [14,116,117]. Numerous studies have shown that targeting CSC surface markers using specific antibodies and small molecules effectively abrogates cancer growth [118]. Targeting not only the surface markers but also intracellular CSC markers, such as ALDH1, Autotaxin, Oct4, Sox2, Nanog, and Hif1a, has been shown many feasibilities to alleviate ovarian CSCs growth and maintenance via small molecule inhibitors or gene knock-down [45,84,86,102,106]. However, many CSC markers have various obstacles to apply on the clinical side due to the limitation of selection and targeting for CSCs.

Targeting CSCs of ovarian cancer has two challenges in clinical application. First, CSC markers cannot explicitly distinguish CSCs from normal stem cells, such as embryonic, hematopoietic, neural, and mesenchymal stem cells. Approximately 73% of currently known CSC surface markers are present on embryonic or adult stem cells [115]. CSC properties, including self-renewal and multi-differentiation potential, are very similar to those of normal stem cells [28,115]. Therefore, these CSC markers are known to originate from cell surface markers of embryonic and adult stem cells [14,116,117,119,120]. In addition, CSCs can arise from the accumulation of epigenetic and genetic alterations in normal stem cells [28]. Some CSC markers have also been detected in normal cells or tissues. Although CD133 is a marker that can be used to distinguish CSCs from various cancer tissues, such as breast, brain, lung, pancreas, liver, prostate, ovary, colon, and head and neck cancers [58], it has also been detected on the surface of differentiated epithelial cells [115]. CD24 has been identified as a CSC marker in a wide range of cancers and has been detected in B cell precursors, neutrophils, neuronal cells, and various epithelial cells [121]. CD117 expression is 24% positive in human embryonic stem cells [122,123]. CD166 is a surface marker in colorectal CSCs and non-small cell lung cancer [124,125]; however, it is also found in many epithelial cells, and it is a marker of human mesenchymal stem cells and intestinal stem cells [126,127]. CD44 is a marker that has been isolated from various solid tumors [128,129], and it is found in human hematopoietic stem cells, MSCs, and adipose-derived stem cells [130,131,132]. Moreover, most intracellular CSC markers, such as Wnt, Notch, Sonic hedgehog, fibroblast growth factor, NANOG, OCT4, SOX2, and MYC, have been detected in normal stem cells. These markers play an essential role in maintaining stem cell properties [98,133,134,135]. CSC markers, such as CD117, CD133, CD24, CD44, OCT4, stage-specific embryonic antigen 4 (SSEA4), leucine-rich repeat-containing G-protein coupled receptor 5 (LGR5), and ALDH1/2 with Ki-67 expression, have been identified in ovarian epithelial cells and cortex regions, although the expression of the markers differed slightly between them [136]. Therefore, the utilization of stem cell markers expressed in normal stem cells or tissues as therapeutic targets against CSCs can lead to the eradication of normal stem cells and disturb normal tissue regeneration or reorganization.

The other challenge is that there is no universal consensus CSC marker to isolate and target all types of CSCs simultaneously. There are multiple types of ovarian CSCs due to intratumoral heterogeneity [137]. The expression of CSC markers, which are already known as potent CSC markers, might be affected according to the differentiation status of CSC and be influenced by splicing variant expression [136,138,139]. For instance, the CD44+/CD24- ovarian cancer cells have CSC-like properties such as tumor-initiating ability and invasion [56]. The CD133+/CD166- ovarian cancer cells possess CSC traits including self-renewal, colony formation, and migration activity [76]. Furthermore, CD133 is expressed on both CSC and differentiated tumor cells but is probably differentially folded as a result of differential glycosylation to mask specific epitopes [140]. CD44-specific variants (v8-v10 isoform) have been reported as a CSC-specific marker in gastric cancer [139,141]. The results suggest that targeting specific surface markers is not enough to eradicate ovarian CSCs due to the heterogeneity of CSCs. Therefore, it is necessary to isolate and verify more CSC-specific functional markers to distinguish CSCs from normal adult stem cells.

### 4.2. Future Directions of CSC-Targeting Therapy

Many researchers have chosen to eradicate CSCs in cancer tissues because these cells are the main factor in building the cancer microenvironment, and they have strongly focused on studying more specific CSC surface markers to distinguish them from those of other cancer or stem cells. However, the application prospect of CSC surface markers is debatable. It is challenging to target a particular CSC surface marker to completely eradicate ovarian CSCs. Therefore, the aforementioned alternative therapy entailed the targeting of at least two CSC markers. For example, CD133 and CD44 aptamer-conjugated dual-targeted nanomicelles loaded with gefitinib, an epidermal growth factor receptor (EGFR) inhibitor, more effectively eradicated CD133/CD44 double-positive lung cancer-initiating cells compared with the single-targeted or non-targeted nanomicelles [142]. Anti-CD24/anti-mesothelin dual-chimeric antigen receptor–natural killer cell (CAR-NK) therapy significantly increased the drug response rate compared with therapy using single-target NK cells. This dual-target strategy was applicable because mesothelin was detected in 42% of ovarian cancer patients [143]. Although the single-target CSC surface marker strategy is sufficient to eliminate ovarian CSCs both in vitro and in vivo, dual marker targeting is more effective in increasing the drug response rate and eradicating ovarian CSCs, including CSCs expressing a single marker. Therefore, dual marker targeting to eliminate CSCs may be a more promising therapeutic method. However, there are many challenges in establishing a therapeutic method and undertaking clinical trials.

Ovarian CSCs are not only involved in the formation of ovarian cancer but constantly interact with the microenvironment around them, including other cells and extracellular components. Notably, cancer-associated fibroblasts, stromal cells, and MSCs in tumor tissues play pivotal roles in cancer malignancy, including cancer growth, invasion, and metastasis. These interactions appear to have positive cross-reactivity through auto- and paracrine signaling. Hence, breaking off the communication between CSCs and the microenvironment, including cancer-associated cells, is essential to eradicate CSCs. The biomarkers that are differentially expressed in normal tissue and cancer-associated cells need to be discovered, and their connection with these interactions should be elucidated. When both cancer cells and adult MSCs were co-inoculated into an immunodeficient mouse model, adult MSCs differentiated into cancer-associated fibroblasts owing to auto- and paracrine factors from cancer cells, and these cells increased the growth and invasion of the cancer cells [144,145]. The ATX–LPA–LPAR1 axis plays an important role in such interactions with cancer cells and adult stem cells in cancer tissues. The inhibition of ATX or LPAR1 using siRNA or small molecules disturbs the differentiation of MSC-derived fibroblasts, and it reduces the metastasis, migration, and cell proliferation of cancer cells [146,147,148,149]. Cancer-associated MSCs induce chemoresistance in ovarian CSCs via platelet-derived growth factor (PDGF) signaling. Combined therapy with sonidegib, a PDGF receptor (PDGFR) inhibitor, and carboplatin broke up the connection between MSCs and ovarian CSCs. The inhibition of PDGFR signaling via sonidegib increased the sensitivity of ovarian CSCs to carboplatin [150]. Therefore, it is likely that the inhibition of both ovarian CSCs and cancer-associated MSCs in the microenvironment is a feasible therapeutic strategy for ovarian cancer (Figure 2).

## 5. Conclusions and Perspectives

CSCs have been implicated in chemoresistance, as well as the tumorigenesis, migration, and invasion of ovarian cancer cells. In patients with ovarian cancer, normal stem cells can be also damaged if CSC-targeted therapy is performed. Therefore, it is necessary to identify more specific markers to selectively eliminate ovarian CSCs. The CSC-targeting antibody can be applied for targeted therapy of cancer via drug conjugation and combination therapy with conventional anti-cancer drugs. Moreover, the CSC-targeting antibody may be useful for development of new technologies targeting the CSCs, including dual-specific antibodies and CAR-T cells. In addition to the direct targeting of CSCs, it will be needed to interfere intercellular communication within the tumor microenvironment to eradicate ovarian CSCs. Moreover, drugs targeting the autotaxin-LPA-LPAR signaling cascade will be highly beneficial for therapy of cancer through abrogating self-renewal and drug-resistant properties of CSCs. Therefore, for effective therapy of ovarian cancer patients, it will be needed to develop CSC-specific antibodies and CSC-targeting therapeutics including chemical drugs, antibody drug conjugates, and CAR-T.

## Figures and Tables

**Figure 1 ijms-22-05059-f001:**
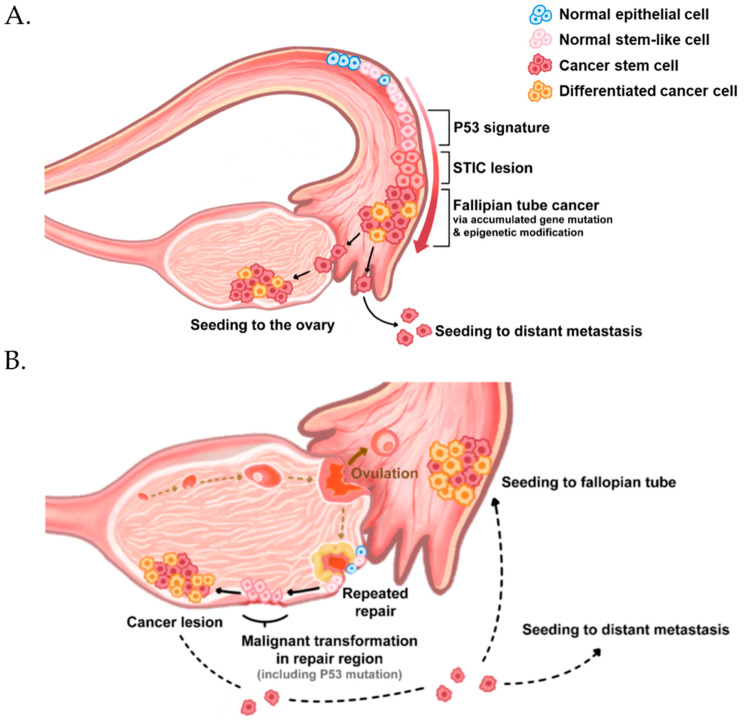
Origin and initiation of high-grade serous ovarian cancer. High-grade serous ovarian cancer originates from the fallopian tube (**A**) and surface epithelia (**B**). Both normal epithelia (blue cells) have a small subpopulation of stem-like cells (Pink cells), which express ALDH, CD117, CD133, CD24, and CD44. They exhibit stem cell properties, such as self-renewal and multi-differentiation potential. (**A**) The stem-like cells in the fallopian tube epithelium acquire p53 mutations owing to uncertain reasons. Then, they expand into serous tubal intraepithelial carcinomas (STICs), transform into fallopian tube cancer, and migrate to seed the ovarian epithelium or distant metastasis. (**B**) In the case of origin from the ovary surface, most researchers believe that the repetitive damages on the ovary surface during ovulation and repair cause accumulated mutations and p53 mutations in the stem-like cells of the ovarian surface epithelium.

**Figure 2 ijms-22-05059-f002:**
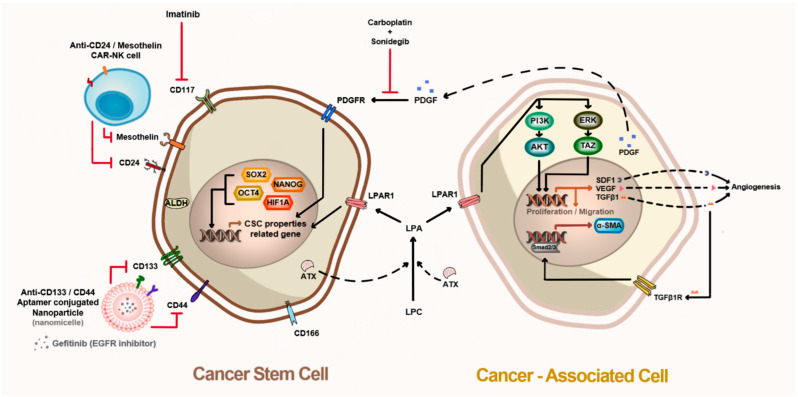
Microenvironment between cancer stem cells and cancer-associated cells in tumor tissue. Cancer stem cells have several surface markers that can be isolated and targeted, such as CD44, CD117, CD24, CD133, and CD166. Cancer stem cells overexpress pluripotent proteins, including ALDH1, ATX, HIF-1α, OCT4, SOX2, and NANOG, which regulate cancer stem cell properties by mediating the expression of surface markers, cytokines, and growth factors. ATX produces LPA, which induces cancer-associated cells to differentiate into myofibroblasts via the TGF-b1–Samd2/3 axis. LPAR1 activation by LPA triggers cell proliferation and migration (homing) toward cancer tissues. Cancer-associated cells secrete PDGF to activate PDGFR in cancer stem cells, thereby maintaining cancer stem cell properties.

**Table 1 ijms-22-05059-t001:** The ovarian cancer stem cell markers and their features.

Protein Names	Features	References
**Cell surface markers of ovarian CSCs**		
CD44	CD44 is the most common CSC marker in the ovary, breast, and pancreas. CD44+ cells have been found in primary, metastatic, and malignant ascites of ovarian carcinoma.	[29,33,34,35,36]
CD117	CD117 has been known as c-kit. CD117 is a receptor tyrosine kinase related to survival, proliferation, migration, tumor progression, and stemness. CD117 is expressed in hematopoietic, mesenchymal, embryonic stem cells. CD117 is involved in stemness and chemoresistance in ovarian cancer.	[19,43,44,45]
CD24	CD24 has been verified as a critical CSC marker in several cancers involving ovarian and colorectal cancers. CD24 is known as the regulation of proliferation, self-renewal, and chemoresistance in ovarian cancer. CD24 positive expressed ovarian cancer patients is poor survival.	[50,51,54,55,56,57]
CD133	CD133 is a glycosylated transmembrane protein and associated with maintaining cancer stemness and tumor metastasis. CD133 has been identified as a prognostic marker in several cancers such as ovarian, glioblastoma, and prostate cancer. CD133 also is correlated with chemoresistance and poor prognosis in ovarian cancer.	[58,59,60,61,62,65]
CD166/ALCAM	CD166 is a transmembrane glycoprotein of the immunoglobulin superfamily. CD166 is overexpressed and is associated with a poor prognosis in various cancers. CD166 induced intracellular cancer stem markers such as OCT4, SOX2, and ALDH1A1. CD166 also is associated with maintaining CSC properties and chemoresistance.	[66,67,68,69,70,71,76]
**Intracellular markers of ovarian CSCs**		
ALDH1	ALDH1 is an enzyme to detoxify reactive aldehydes. ALDH composes three classes. ALDH1 consists of three isozymes; ALDH1A1 is predominantly a CSC marker. ALDH elevated stem-cell marker expression, self-renewal, proliferation, and differentiation potential. ALDH is also associated with invasion, drug resistance, and worse progression-free survival in ovarian cancer.	[78,79,80,81,82,83,84,85,86]
Autotaxin	Autotaxin possesses lysophospholipase D activity and produces lysophosphatidic acid (LPA). Autotaxin is expressed in several tissues, including body fluid, and it is highly expressed in the breast, glioblastoma, prostate, hepatocellular carcinoma, and melanoma. Autotaxin enhances migration, sphere formation, and cancer stemness markers.	[87,88,89,90,91,92,93,94,95]
Pluripotency-associated factors	OCT4, SOX2, and NANOG are the most common intracellular cancer stem markers in normal and cancer stem cells. Among these factors, SOX2 plays a more substantial role in early tumor initiation and tumor relapse in ovarian cancer, including breast, gastric, and lung. The knock-down SOX2 inhibits ovarian CSC survival and spheroid formation.	[12,97,98,101,102]
HIF-1α	Hif1a is one of the hypoxia-inducible factors and master regulators of hypoxia. Hif1a is correlated with growth, survival, and metastasis during hypoxia conditions. Hif1a can elevate the expression of pluripotent stem markers and plays an essential role in maintaining CSC properties.	[102,103,104,105,107]

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
