# Peer review of "Therapeutic Strategies for Targeting Ovarian Cancer Stem Cells"

_ijms, 2021, doi:10.3390/ijms22105059_

Round 1

Reviewer 1 Report

This is an excellent review with a great organization that makes comprehensive consideration of what is currently know about cancer stem cells in ovarian cancer.  The writing is of the highest caliber.  On line 43 change "of them" to "of these patients".  On line 92 change "epithelium" to "epithelia".

I enjoyed reviewing this excellent review.

Author Response

Reviewer 1

This is an excellent review with a great organization that makes comprehensive consideration of what is currently know about cancer stem cells in ovarian cancer.  The writing is of the highest caliber.  On line 43 change "of them" to "of these patients".  On line 92 change "epithelium" to "epithelia".

Answer: I highly appreciate your affirmative comments to improve the paper. According to your comments, I have changed the appropriate terms on line 43 and 92.

Reviewer 2 Report

This manuscript is a nice summary of the CSCs in ovarian cancer. The manuscript could be a valuable resource for the readers to gain a comprehensive understanding of CSCs in ovarian cancer. In addition, there are a few comments, as noted below, recommended to the authors to make this manuscript more valuable to the reader. 

  1. Figure 1 is not a good representation to show the development of HGSC in the fallopian tube (FT). The p53 sig. is not well presented. And the overall path drawn is very confusion and not well presented. it is recommended that authors consider improving this figure.
  2. Althoug the authors have presented a commentary on various aspects of CSCs in ovarian cancer as separate sections however, an overall commentary/discussion is missing. It would be good to add a discussion section in which the authors could bring together their perspective and the learnings from all sections on CSCs. It would be good to highlight some of the limitations and challenges, overall, in the field of CSCs in ovarian cancer. 

    It would be great to include a short discussion on the current models/techniques used by the researchers to study CSCs in vitro and their limitations. 

    Authors should also discuss some future directions, which would  add value to this manuscript. 

    A table listing all markers discussed in the manuscript, along with their references and key features would be a great value to the readers. 

Author Response

Reviewer 2

This manuscript is a nice summary of the CSCs in ovarian cancer. The manuscript could be a valuable resource for the readers to gain a comprehensive understanding of CSCs in ovarian cancer. In addition, there are a few comments, as noted below, recommended to the authors to make this manuscript more valuable to the reader. 

  1. Figure 1 is not a good representation to show the development of HGSC in the fallopian tube (FT). The p53 sig. is not well presented. And the overall path drawn is very confusion and not well presented. it is recommended that authors consider improving this figure.

Answer: Thank you for the valuable comment. We have separated the original figure 1 to two diagrams to avoid misunderstanding and obscurity. In the new figure 1A, the fallopian epithelia-originated serous adenocarcinoma was displayed, and ovarian epithelia-originated serous adenocarcinoma was shown in figure 1B.

2, Although the authors have presented a commentary on various aspects of CSCs in ovarian cancer as separate sections however, an overall commentary/discussion is missing. It would be good to add a discussion section in which the authors could bring together their perspective and the learnings from all sections on CSCs. It would be good to highlight some of the limitations and challenges, overall, in the field of CSCs in ovarian cancer. 

Answer: Thank you for the helpful comment. According to the referee’s comment, we added a paragraph describing perspectives in “Conclusion and perspectives” section as follows.

Lines 389-402: “CSCs have been implicated in chemoresistance, as well as the tumorigenesis, migration, and invasion of ovarian cancer cells. In patients with ovarian cancer, normal stem cells can be also damaged if CSC-targeted therapy is performed. Therefore, it is necessary to identify more specific markers to selectively eliminate ovarian CSCs. The CSC-targeting antibody can be applied for targeted therapy of cancer via drug conjugation and combination therapy with conventional anti-cancer drugs. Moreover, the CSC-targeting antibody may be useful for development of new technologies targeting the CSCs, including dual-specific antibodies and CAR-T cells. In addition to the direct targeting of CSCs, it will be needed to interfere intercellular communication within the tumor microenvironment to eradicate ovarian CSCs. Moreover, drugs targeting the autotaxin-LPA-LPAR signaling cascade will be highly beneficial for therapy of cancer through abrogating self-renewal and drug resistant properties of CSCs. Therefore, for effective therapy of ovarian cancer patients, it will be needed to develop CSC-specific antibodies and CSC-targeting therapeutics including chemical drugs, antibody drug conjugates, and CAR-T.”

  1. It would be great to include a short discussion on the current models/techniques used by the researchers to study CSCs in vitro and their limitations. 

Answer: Thank you for the nice comment. According to the referee’s comment, we added a paragraph describing the current research techniques for targeting CSCs in section 4.2 on lines 278-291. It includes the usage of CSC markers for targeting CSCs, the importance of CSC surface markers, the CSC isolation, and targeting in the experimental field. Their limitation was described on lines 292-335 as follows.

Lines 278-291: “CSC surface markers are more accessible to therapeutic antibodies and small molecules than other CSC markers; thus, it is a feasible target that can be utilized to treat various cancers [116]. Moreover, considering the function of CSC markers in maintaining CSC properties, it is straightforward to isolate specific CSCs and use the diagnostic markers for predicting patient survival. Although there are several debates on the isolation and targeting of CSC surface markers, many researchers believe that eliminating specific CSCs can eradicate whole cancer tissues [14, 117, 118]. Numerous studies have shown that targeting CSC surface markers using specific antibodies and small molecules effectively abrogates cancer growth [116, 119]. Targeting not only the surface markers but also intracellular CSC markers, such as ALDH1, Autotaxin, Oct4, Sox2, Nanog and Hif1a, has been shown many feasibilities to alleviate ovarian CSCs growth and maintenance via small molecule inhibitors or gene knock-down [45, 84, 86, 102, 106]. However, many CSC markers have various obstacles to apply on the clinical side due to the limitation of selection and targeting for CSCs.“

Lines 292-335: “Targeting CSCs of ovarian cancer has two challenges in clinical application. First, CSC markers cannot explicitly distinguish CSCs from normal stem cells, such as embryonic, hematopoietic, neural, and mesenchymal stem cells. Approximately 73% of currently known CSC surface markers are present on embryonic or adult stem cells [116]. CSC properties, including self-renewal and multi-differentiation potential, are very similar to those of normal stem cells [28, 116]. Therefore, these CSC markers are known to originate from cell surface markers of embryonic and adult stem cells [14, 117, 118, 120, 121]. In addition, CSCs can arise from the accumulation of epigenetic and genetic alterations in normal stem cells [28]. Some CSC markers have also been detected in normal cells or tissues. Although CD133 is a marker that can be used to distinguish CSCs from various cancer tissues, such as breast, brain, lung, pancreas, liver, prostate, ovary, colon, and head and neck cancers [59], it has also been detected on the surface of differentiated epithelial cells [116]. CD24 has been identified as a CSC marker in a wide range of cancers and has been detected in B cell precursors, neutrophils, neuronal cells, and various epithelial cells [122]. CD117 expression is 24% positive in human embryonic stem cells (36, 70). CD166 is a surface marker in colorectal CSCs and non-small cell lung cancer (129, 9); however, it is also found in many epithelial cells, and is a marker of human mesenchymal stem cells and intestinal stem cells (127, 128). CD44 is a marker that has been isolated from various solid tumors (131, 134), and is found in human hematopoietic stem cells, MSCs, and adipose-derived stem cells (132, 91, 133). Moreover, most intracellular CSC markers, such as Wnt, Notch, Sonic hedgehog, fibroblast growth factor, NANOG, OCT4, SOX2, and MYC, have been detected in normal stem cells. These markers play an essential role in maintaining stem cell properties (154, 155, 158, 159). CSC markers, such as CD117, CD133, CD24, CD44, OCT4, stage-specific embryonic antigen 4 (SSEA4), leucine-rich repeat-containing G-protein coupled receptor 5 (LGR5), and ALDH1/2 with Ki-67 expression, have been identified in ovarian epithelial cells and cortex regions, though the expression of the markers differed slightly between them [123]. Therefore, the utilization of stem cell markers expressed in normal stem cells or tissues as therapeutic targets against CSCs can lead to the eradication of normal stem cells and disturb normal tissue regeneration or reorganization.

The other challenge is that there is no universal consensus CSC marker to isolate and target all types of CSCs simultaneously. There are multi-types of ovarian CSCs due to intratumoral heterogeneity [124]. Expression of CSC markers, which are already known as potent CSC markers, might be affected according to the differentiation status of CSC and be influenced by splicing variant expression [123, 125, 126]. For instance, the CD44+/CD24- ovarian cancer cells have CSC-like properties such as tumor-initiating ability and invasion [57]. The CD133+/CD166- ovarian cancer cells possess CSC traits including self-renewal, colony formation, and migration activity [77]. Furthermore, CD133 is expressed on both CSC and differentiated tumor cells, but is probably differentially folded as a result of differential glycosylation to mask specific epitopes [127]. CD44 specific variants (v8-v10 isoform) has been reported as a CSC-specific marker in gastric cancer [126, 128]. There result suggest that targeting specific surface markers is not enough to eradicate ovarian CSCs due to the heterogeneity of CSCs. Therefore, it is necessary to isolate and verify more CSC-specific functional markers to distinguish CSCs them from normal adult stem cells.”

  1. Authors should also discuss some future directions, which would add value to this manuscript. 

Answer: Thank you for the valuable comment. According to the comment, we have described future directions to overcome the current limitations of CSC markers-targeted therapy in section 4.3 (section title: Future directions of CSC-targeting therapy). The future directions include targeting multiple CSC markers using a dual-specific antibody, CSC marker-specific CAR-NK cells (lines 337-355), and disconnecting between CSCs and their microenvironment (lines 356-378) as follows.

Lines 337-355: “Many researchers have chosen to eradicate CSCs in cancer tissues because these cells are the main factor in building the cancer microenvironment, and they have strongly focused on studying more specific CSC surface markers to distinguish them from those of other cancer or stem cells. However, the application prospect of CSC surface markers is debatable. It is challenging to target a particular CSC surface marker to completely eradicate ovarian CSCs. Therefore, the aforementioned alternative therapy entailed the targeting of at least two CSC markers. For example, CD133 and CD44 aptamer-conjugated dual-targeted nanomicelles loaded with gefitinib, an epidermal growth factor receptor (EGFR) inhibitor, more effectively eradicated CD133/CD44 double-positive lung cancer-initiating cells compared with the single-targeted or non-targeted nanomicelles [129]. Anti-CD24/anti-mesothelin dual-chimeric antigen receptor–natural killer cell (CAR-NK) therapy significantly increased the drug response rate compared with therapy using single-target NK cells. This dual-target strategy was applicable because mesothelin was detected in 42% of ovarian cancer patients [130]. Although the single-target CSC surface marker strategy is sufficient to eliminate ovarian CSCs both in vitro and in vivo, dual marker targeting is more effective in increasing the drug response rate and eradicating ovarian CSCs, including CSCs expressing a single marker. Therefore, dual marker targeting to eliminate CSCs may be a more promising therapeutic method. However, there are many challenges in establishing a therapeutic method and undertaking clinical trials.”

Lines 356-378: “Ovarian CSCs are not only involved in the formation of ovarian cancer but constantly interact with the microenvironment around them, including other cells and extracellular components. Notably, cancer-associated fibroblasts, stromal cells, and MSCs in tumor tissues play pivotal roles in cancer malignancy, including cancer growth, invasion, and metastasis. These interactions appear to have positive cross-reactivity through auto- and paracrine signaling. Hence, breaking off the communication between CSCs and the microenvironment, including cancer-associated cells, is essential to eradicate CSCs. The biomarkers that are differentially expressed in normal tissue and cancer-associated cells need to be discovered, and their connection with these interactions should be elucidated. When both cancer cells and adult MSCs were co-inoculated into an immunodeficient mouse model, adult MSCs differentiated into cancer-associated fibroblasts owing to auto- and paracrine factors from cancer cells, and these cells increased the growth and invasion of the cancer cells [131, 132]. The ATX–LPA–LPAR1 axis plays an important role in such interactions with cancer cells and adult stem cells in cancer tissues. The inhibition of ATX or LPAR1 using siRNA or small molecules disturbs the differentiation of MSC-derived fibroblasts, and reduces the metastasis, migration, and cell proliferation of cancer cells [133-136]. Cancer-associated MSCs induce chemoresistance in ovarian CSCs via platelet-derived growth factor (PDGF) signaling. Combined therapy with sonidegib, a PDGF receptor (PDGFR) inhibitor, and carboplatin broke up the connection between MSCs and ovarian CSCs. The inhibition of PDGFR signaling via sonidegib increased the sensitivity of ovarian CSCs to carboplatin [137]. Therefore, it is likely that the inhibition of both ovarian CSCs and cancer-associated MSCs in the microenvironment is a feasible therapeutic strategy for ovarian cancer (Figure 2).”

  1. A table listing all markers discussed in the manuscript, along with their references and key features would be a great value to the readers. 

Answer: Thank you the valuable comment. According to the comment, we have added Table 1 which include ovarian cancer stem markers, their brief features, and adequate references.